# Effects of Capsular Polysaccharide amount on Pneumococcal-Host interactions

**Jiaqi Zhu[1‡], Annie R. Abruzzo[1‡], Cindy Wu[1], Gavyn Chern Wei Bee[1], Alejandro Pironti[1,2], Gregory Putzel[1,2], Surya D. Aggarwal[1], Hannes Eichner[1,3], Jeffrey N. Weiser[1]***

**1** Department of Microbiology, New York University Grossman School of Medicine, New York, New York, United States of America, **2** Microbial Computational Genomic Core Lab, Department of Microbiology, New York University Grossman School of Medicine, New York, New York, United States of America, **3** Department of Microbiology, Tumor and Cell Biology, Karolinska Institutet, and Clinical Microbiology, Bioclinicum, Karolinska University Hospital Solna, Solna, Sweden

‡ These authors are co-first authors on this work.
* jeffrey.weiser@nyulangone.org

**Data Availability Statement:** The link to our data Dryad provided is: https://datadryad.org/stash/share/jpNBGQpWrHUawsaKV3Gen_3qFoVifuW5JqxMX-1xVd8. Nucleotide sequencing

## Abstract

Among the many oral streptococci, *Streptococcus pneumoniae* (*Spn*) stands out for the capacity of encapsulated strains to cause invasive infection. Spread beyond upper airways, however, is a biological dead end for the organism, raising the question of the benefits of expending energy to coat its surface in a thick layer of capsular polysaccharide (CPS). In this study, we compare mutants of two serotypes expressing different amounts of CPS and test these in murine models of colonization, invasion infection and transmission. Our analysis of the effect of CPS amount shows that *Spn* expresses a capsule of sufficient thickness to shield its surface from the deposition of complement and binding of antibody to underlying epitopes. While effective shielding is permissive for invasive infection, its primary contribution to the organism appears to be in the dynamics of colonization. A thicker capsule increases bacterial retention in the nasopharynx, the first event in colonization, and also impedes IL-17-dependent clearance during late colonization. Enhanced colonization is associated with increased opportunity for host-to-host transmission. Additionally, we document substantial differences in CPS amount among clinical isolates of three common serotypes. Together, our findings show that CPS amount is highly variable among *Spn* and could be an independent determinant affecting host interactions.

## Author summary

Many leading bacterial pathogens are coated by a thick layer of polysaccharide, which is referred to as a capsule. This capsular layer shields the underlying surface of the organism from recognition by host factors like antibodies that would otherwise facilitate clearance during invasive infection. Pathogens with capsules such as *Streptococcus pneumoniae*, however, generally reside on mucosal surfaces (colonization) without causing disease. Thus, the benefit to the organism of expressing a thick capsule is unclear. This study uses genetically-modified strains of *S. pneumoniae* to test how capsule thickness (measured by the amount of capsular polysaccharide) affects host-pathogen interactions using infant

data has been uploaded to NCBI under BioProject PRJNA930766.

**Funding:** This manuscript was supported by NIH grants awarded to JNW (RO1 AI50893, R37 AI38446 and R21 AI50867).The funders had no role in study design, data collection and analysis, decision to publish, or preparation of the manuscript. No authors received a salary from any funders.

**Competing interests:** The authors have declared that no competing interests exist.

mouse models. We show that thickness sufficient to protect the organism during invasive infection also benefits the organism during colonization. This effect is attributed to increased retention of the organism when it first encounters the mucosal surface of the upper respiratory tract. Additionally, we demonstrate that capsule thickness is a highly variable trait among *S. pneumoniae* clinical isolates. Taken together, our findings could account for differences in the ability of *S. pneumoniae* isolates to both colonize the host upper respiratory tract and cause invasive infection.

## Introduction

*Streptococcus pneumoniae* (*Spn*, the pneumococcus) is the leading bacterial pathogen associated with deaths in children less than five years of age worldwide [1]. Virulent *Spn* are covered by a thick layer of capsular polysaccharide (CPS) [2]. In the absence of this layer, the organism is unable to cause the main invasive infections (pneumonia, sepsis, meningitis) responsible for its burden on public health [3]. The species expresses at least 100 'types' of structurally and immunologically distinct CPSs. While CPS is the basis of all currently licensed *Spn* vaccines, only limited CPS types are included as antigens. Thus, the heterogeneity of *Spn* capsule limits the effectiveness of current prevention strategies.

Because of its importance as a determinant of virulence and vaccine antigen, there has been much focus on understanding the contribution of CPS to the biology of the pneumococcus [4]. These studies have emphasized the role of capsule during invasive infections. Capsules are generally considered to be antiphagocytic, since they inhibit the deposition of opsonizing complement and antibody onto the bacterial surface [5]. For *Spn*, different CPS types vary in their ability to block opsonophagocytic clearance, which is facilitated once CPS type-specific antibody is generated [6]. Invasive infection, however, is a biological 'dead end' for *Spn*, whose lifecycle depends on colonization of the upper respiratory tract (URT) of humans (the carrier state) and transmission from one carrier to another.

The biosynthesis of a thick capsule coat would be expected to confer a substantial metabolic burden in proportion to the amount of CPS expressed, which must be offset by a biological advantage. *In vivo* studies suggest that capsules enhance *Spn* retention along the URT epithelium by limiting bacterial binding to mucus and, thereby, preventing mechanical clearance via mucociliary flow [7]. However, many isolates of *Spn* and other members of the diverse *Streptococcus mitis* group are nonencapsulated [8], yet successfully colonize the upper respiratory tract (URT) as commensals [9–11]. Capsule also inhibits *Spn* adherence to epithelial cells cultured *in vitro*, raising the question of how encapsulated organisms persist on mucosal surfaces [12–15].

It has been suggested that many encapsulated species deal with these potential advantages and disadvantages of capsule by variation between forms with high and low levels of CPS expression. For many *Spn*, this variation may be observed as on-off switching or phase variation between opaque and transparent colony forms, which express thick and thin capsules, respectively, and differ both in their abilities to colonize and cause invasive infection [16]. A molecular switch involving spontaneous recombination events affecting genome-wide methylation is now known to control *Spn* phase variation [17–19]. However, because this mechanism affects many bacterial characteristics besides CPS amount, phase variation in opacity is not a specific means to determine the consequences of CPS amount diversity. There is also evidence that capsule expression is dynamic within the respiratory tract [13] and that it may shed in response to the epithelium [20].

The purpose of this study is to use defined mutants differing only in the amount of CPS to reassess the specific contributions of capsule in *Spn*-host interactions. These are tested in an infant mouse model that recapitulates key aspects of early childhood infection, including prolonged carriage, direct transit from the mucosal surface of the URT to bloodstream (occult bacteremia), bacteremic infection, and host-to-host transmission.

## Results

### Characterization of strains expressing different amounts of CPS

A type 6A clinical isolate (P376) was screened to identify spontaneous mutants expressing different amounts of CPS based on altered colony morphology (Table 1). Two candidates, P384 and P385, previously shown to express ~45% and ~40%, respectively, of WT levels of type 6A CPS using a quantitative capture ELISA were further analyzed by whole genome sequencing [21]. In comparison to the parental strain, both strains contained a missense mutation in the *cps* sugar transferase gene *cpsE*. In addition, both strains contained a second point missense mutation in an unrelated locus. To confirm that only the former mutation was responsible for altered CPS expression, the WT *cps* region was used to correct the *cpsE* point mutation in T6A$^{40\%}$, resulting in a corrected mutant (T6A$^{CM}$), identified among transformants by screening for WT colony morphology and validated by sequencing. After confirming restoration of full CPS expression by immunoblotting (Fig 1A), a functional assay was used to compare the relative ability of the strains to shield the bacterial surface from antibody binding to a conserved epitope underlying the capsule. As expected, strains with higher CPS expression were more protected from the binding of mAb TEPC-15 to phosphorylcholine, a feature of *Spn* teichoic acids, as measured by flow cytometry (Fig 1C). Differences in surface shielding between T6A$^{100\%}$ and T6A$^{40\%}$ were similar when grown *in vitro* or obtained from URT lavages of colonized pups and analyzed *ex vivo*.

Our analysis also included a previously described set of strains expressing different amounts of the type 4 CPS [22]. An unencapsulated mutant (P2422) containing the Sweet Janus cassette inserted in *cps* was used as recipient to restore the WT locus and to generate constructs with modified *cps* protomer regions expressing ~20% or ~50% of WT levels of type 4 CPS. We confirmed differences in type 4 CPS expression by immunoblotting (Fig 1B) and then compared

**Table 1. Bacterial strains.**

| Strain Designation | Strain number | Description | Reference |
|---|---|---|---|
| T4Sm$^R$ | P2406 | Streptomycin resistant derivative of TIGR4 (T4 WT) | [42] |
| T4$^{0\%}$ | P2422 | P2406 with Sweet Janus cassette replacing *cps* locus | [22] |
| T4$^{100\%}$ | P2438 | P2422 transformed with P2406 genomic DNA to restore capsule expression | [22] |
| T4$^{20\%}$ | P2492 | P2406 mutated to carry heterologous *cps* promoter. Expresses ~20% of WT levels of T4 CPS. | [22] |
| T4$^{50\%}$ | P2480 | P2406 with pCAT mutation in the *cps* promoter. Expresses ~50% of WT levels of T4 CPS | [22] |
| T6A$^{100\%}$ | P376 | T6A clinical isolate | [21] |
| T6A$^{45\%}$ | P384 | Spontaneous variant of P376. Expresses ~45% of WT levels of T6A CPS. Contains two missense mutations: CiaR$_{Asp10Glu}$ and CpsE$_{Val211Asp}$. | [21] |
| T6A$^{40\%}$ | P385 | Spontaneous variant of P376. Expresses ~40% of WT levels of T6A CPS. Contains two missense mutations: CpsE$_{Phe297Ile}$ and YqfR$_{Val314Ile}$ | This study |
| T6A$^{CM}$ | P2752 | Corrected mutant of P385 by transformation of the T6A *cps* locus to restore the WT *cpsE* | This study |
| - | P592 | Clinical isolate, type 6A | Collection of R. Austrian |
| - | P2797 | Variant of P592 containing CpsE$_{Thr321Ala}$ | This study |

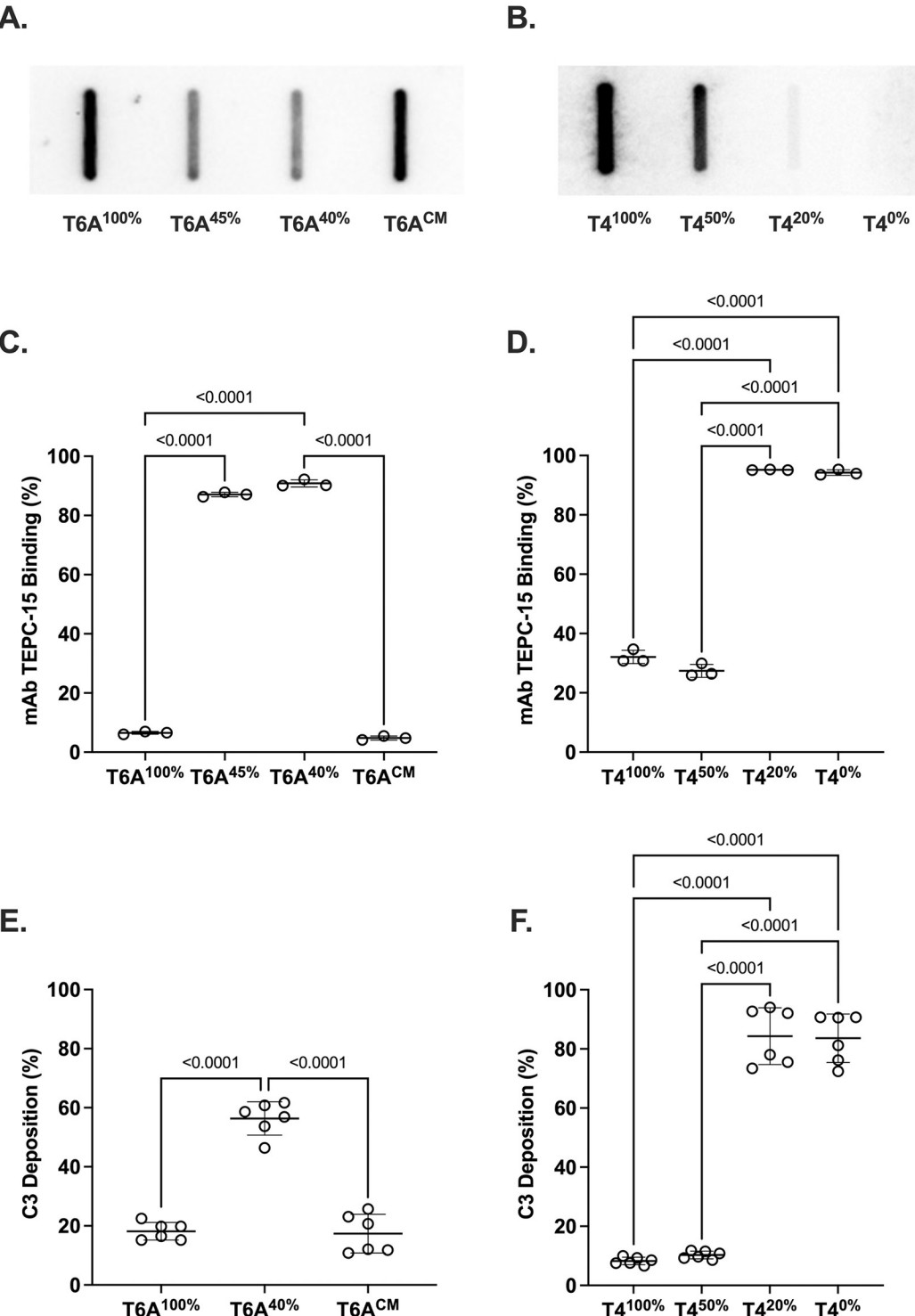

**Fig 1. Characterization of *Spn* strains expressing different amounts of capsular polysaccharide phosphorylcholine.**
Immunoblots of *Spn* lysates of serotype (A) T6A or (B) T4 strains with serotype-specific anti-capsule antibody. (C-F) Shielding effect of the capsule evaluated using quantification by flow cytometry. Binding of mAb TEPC-15 to phosphorylcholine on (C) T6A or (D) T4 strains. Deposition of complement (C3b) from normal mouse serum using anti-mouse C3 antibody on (E) T6A or (F) T4 strains. Statistical significance based on the percentage of *Spn* cells in which antibody binding was detected was determined using the Kruskal-Wallis test followed by Dunn's correction. Corrected mutant (CM).

their ability to shield the bacterial surface from binding of mAb TEPC-15. Strains expressing 0% or 20% of CPS exhibited no shielding, whereas strains expressing 50% or 100% were equivalently shielded (Fig 1D). This suggests a threshold for effective shielding between 20% and 50% of WT levels of T4 CPS.

## Role of CPS amount in invasive infection

The virulence of the strains was then compared following intraperitoneal (IP) challenge of a low dose (100 CFU) in 10-day old mice. For the type 6A strains, the humane endpoint was reached by 24 h post-inoculation (p.i.) for all pups given T6A$^{100\%}$ or T6A$^{CM}$ and blood cultures showed high levels of bacteremia (Fig 2A). In contrast, none of the pups challenged with the mutants expressing T6A$^{40\%}$ appeared septic or had detectable bacteremia. For the type 4 constructs, bacteremia was only detected in pups challenged with the fully encapsulated strain (Fig 2B). These results demonstrate the contribution of capsule to virulence in this model and show a relationship between amounts of CPS and the capacity to sustain bloodstream infection.

Because mice are naïve hosts and lack naturally-acquired anti-*Spn* antibodies, complement is the main serum opsonin. Complement deposition by the strains was compared using flow cytometry by detection of C3b bound to the bacterial surface following incubation with normal mouse serum. Similar to the binding of mAb TEPC-15, lower CPS expression (e.g. T6A$^{40\%}$ and T6A$^{45\%}$) was associated with increased C3b deposition (Fig 1E). (Subsequently, further analysis of T6A$^{45\%}$ was discontinued due to the established *in vivo* role of *ciaR* and the potential confounding effects of the point mutation in this gene [23]) The pattern of C3b deposition for the type 4 constructs paralleled mAb TEPC-15 binding with a threshold between 20% and 50% of WT levels of CPS (Fig 1F). Thus, the amount of CPS correlated with inhibition of both cell surface antibody binding and complement deposition.

To confirm that complement deposition was responsible for the attenuated virulence of mutants with lower amounts of CPS, the IP challenge was repeated in pups in which serum complement was depleted by prior treatment with cobra venom factor. In pups lacking complement activity, strains T6A$^{40\%}$ and T4$^{50\%}$ were able to cause high level bacteremia unlike in complement-sufficient mice (Fig 2C and 2D). Together these findings demonstrate that only by the expression of high levels of CPS is *Spn* able to shield its surface from opsonins that would otherwise promote its clearance during bloodstream infection.

## Role of CPS amount in colonization dynamics

We then considered how amounts of CPS affect colonization. Mice at 10 (type 6A) or 4 (type 4) days of life were given an intranasal (IN) inoculum ($10^2$ CFU), and the numbers of colonizing *Spn* were assessed in quantitative cultures of nasal lavages over a 28-day period (Fig 3A). At the initial time point (day 1 p.i.), the density of colonizing *Spn* was significantly lower for the T6A$^{40\%}$ compared to both the parent and corrected mutant. This difference was maintained throughout the 28-day time course, by the end of which most mice cleared the less encapsulated mutant. For the type 4 constructs, colonization density generally correlated with amounts of CPS over the time course and was significantly reduced for the strains expressing 0% and 20% of WT levels of CPS (Fig 3C). Only these two constructs were completely cleared by the final time point.

The dynamics of colonization density, therefore, showed two distinct effects of CPS amount for both type 6A and 4 strains. In the early phase (day 1), differences in colonization density correlated with CPS amount, and this effect persisted through 14 days p.i. In later phase, there was a more precipitous decline in colonization density from 14-day p.i. for the T6A and T4 strains expressing less CPS that ultimately led to clearance.

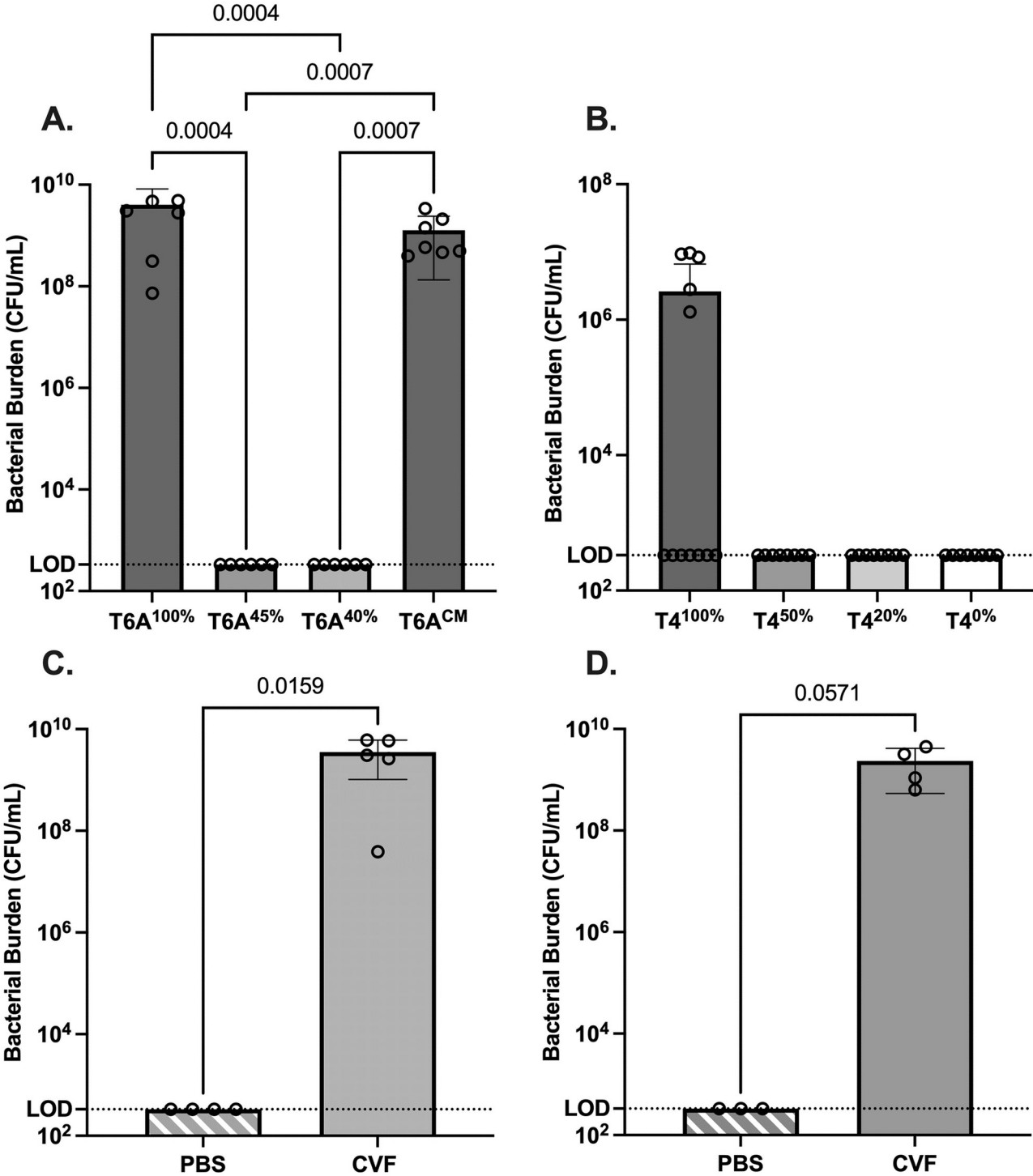

**Fig 2. Effect of differences in CPS amount on invasive infection.** (A-B) 10-day-old pups were challenged intraperitoneally with *Spn* ($\sim 10^2$ CFU) of (A) T6A or (B) T4 strains and sacrificed 24 hrs later when blood was collected for quantitative culture to determine bacterial burden in the bloodstream. Statistical significance was determined using a Kruskal-Wallis test followed by Dunn's correction. (C-D) Pups were pretreated with either cobra venom factor (CVF) or vehicle control (PBS). Limit of detection (LOD). Statistical significance was determined using the Mann-Whitney test.

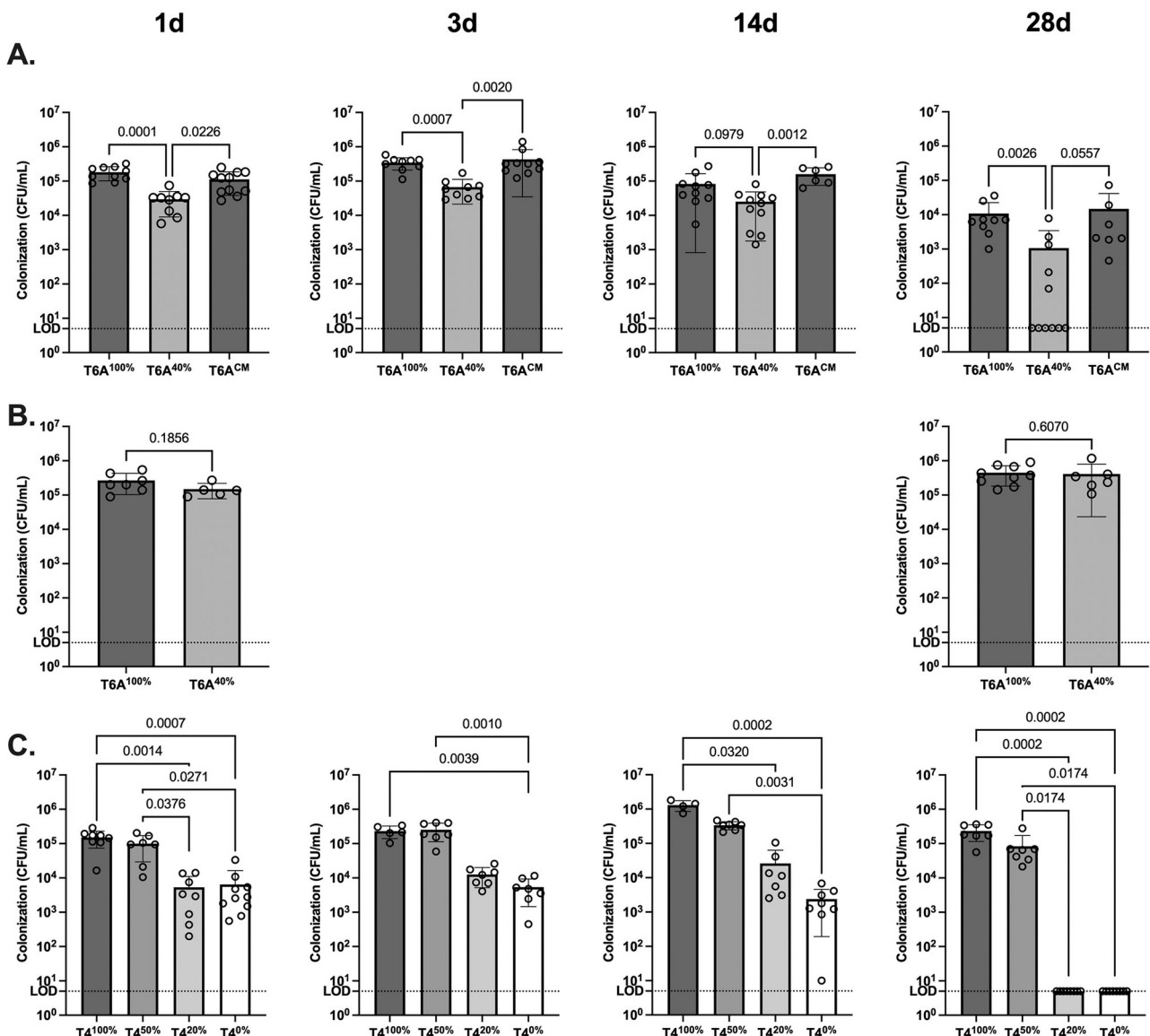

**Fig 3. Effect of differences in CPS amount on colonization dynamics.** (A) 10-day-old pups (C57BL6) were challenged intranasally (IN) with T6A strains (~$10^2$ CFU) and sacrificed at 1 day, 3 days, 14 days, or 28 days post-infection. Upper respiratory tract (URT) lavages were collected for quantification of colonization density. Statistical significance was determined using a Kruskal-Wallis test followed by Dunn's correction. (B) *IL17ra⁻/⁻* pups at 10 days of age were challenged IN with T6A strains (~$10^2$ CFU) and sacrificed at 1 day or 28 days post-infection. URT lavages were collected for quantification of colonization density. Statistical significance was determined using the Mann-Whitney test. (C) C57BL6 pups at 4 days of age were challenged IN with T4 strains (~$10^2$ CFU) and sacrificed at 1 day, 3 days, 14 days, or 28 days post-infection. URT lavages were collected for quantification of colonization densities. Statistical significance was determined using a Kruskal-Wallis test followed by Dunn's correction. Limit of detection (LOD).

To further examine the early effect, we measured colonization density at 4 hrs p.i. Prior studies established that this timepoint is before bacterial replication becomes the main factor driving bacterial numbers, and, therefore, CFU counts largely reflect the retention of the inoculum in the URT [24]. Retention correlated with the amount of CPS and was significantly impaired for the T6A⁴⁰% and T4⁰% mutants compared to more fully encapsulated strains of the same type (Fig 4A and 4B). To control for potential host-related differences, the 4-hour

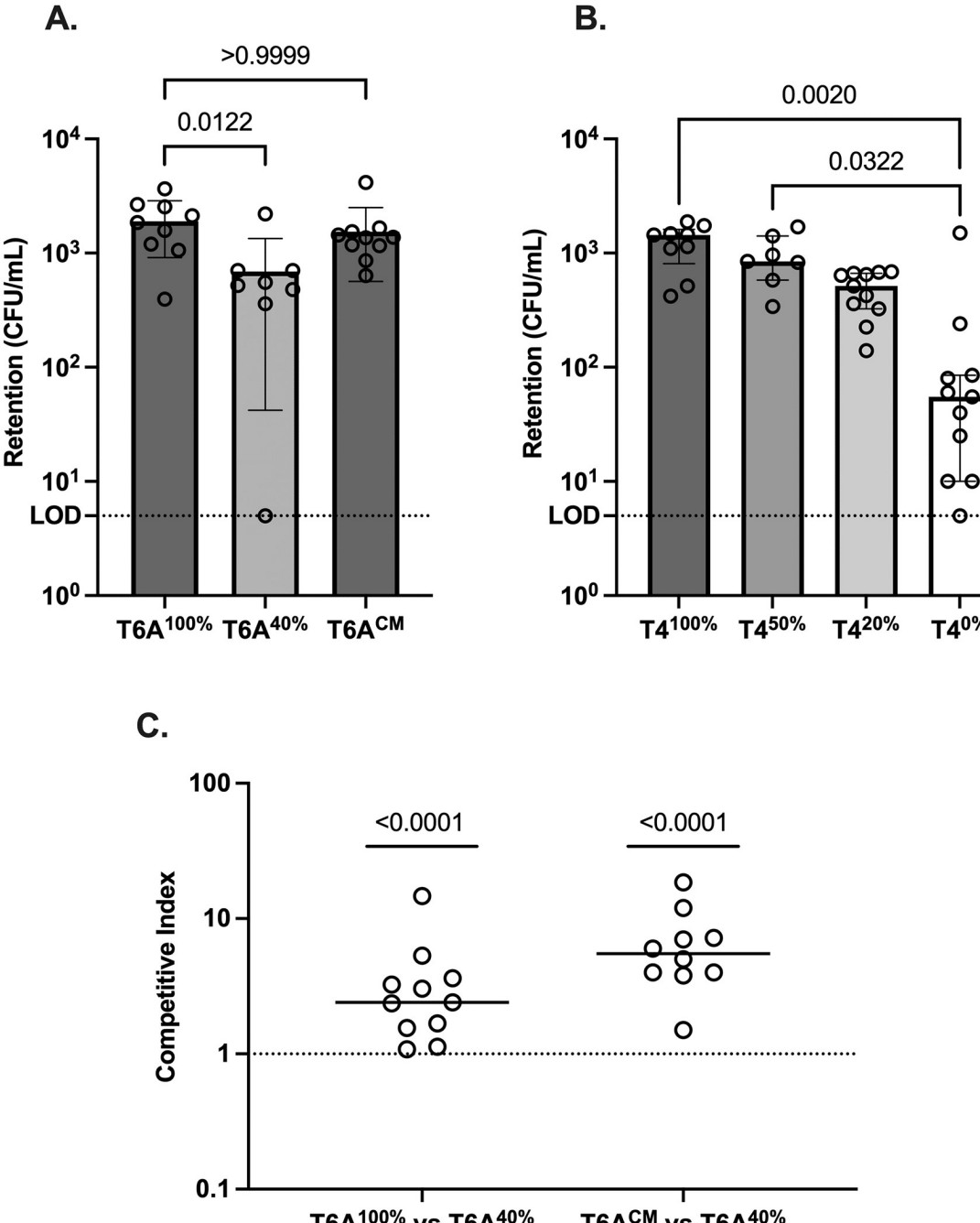

**Fig 4. Effect of differences in CPS amount on retention of the inoculum.** Pups at 10 (T6A) or 5 (T4) days of age were challenged intranasally with *Spn* (~$10^{2-3}$ CFU) strains (A) T6A or (B) T4. The pups were sacrificed 4 hours after infection to obtain URT lavages for quantitative culture. Limit of detection (LOD). Statistical significance was determined using Kruskal-Wallis test followed by Dunn's correction. (C) Competition assay of T6A strains. Pups at 5–10 days of age were challenged IN (~$10^{2-3}$ CFU/strain) with an equal mixture of the two strains indicated below and sacrificed 4 hours later. URT lavages were collected for quantitative culture with strains distinguished by colony morphology. Values represent the ratio of strains in the lavage (output) over the ratio in the inoculum (input) for each pup. Statistical significance was determined using the Mann-Whitney test in comparison to a hypothetical value of 1 (representing no competition).

retention assay was repeated in a competitive assay in which equivalent numbers of T6A[100%] or T6A[CM] competed with T6A[40%] were delivered to the same pup (Fig 4C). Both fully encapsulated strains outcompeted the point mutant expressing less CPS. These results demonstrate that the amount of CPS affects initial interaction of *Spn* and the host.

To further examine the later effect, we considered host pathways known to mediate clearance of colonization. Prior studies showed this requires IL-17-dependent recruitment of professional phagocytes, as mice lacking the common receptor for IL-17 cytokines exhibit impaired phagocyte recruitment and delayed clearance [25, 26]. The more accelerated clearance in WT mice of T6A[40%] was absent in *IL-17ra[-/-]* mice (Fig 3B). This observation shows that amounts of CPS affect IL-17-dependent clearance mechanisms.

## Role of CPS amount in other host interactions

The type 6A isolate used in this study is capable of causing bacteremia and sepsis following URT colonization (occult bacteremia) [27]. Thus, by comparing survival following IN challenge, we could assess the overall effects of amounts of CPS, since this outcome involves the steps of 1) retention of *Spn* in the URT, 2) establishment of colonization, 3) transit from the mucosal surface to bloodstream, and 4) proliferation in the bloodstream. When tested in 10-day old mice, the humane endpoint was reached beginning 2 days p.i. (Fig 5). The point mutant expressing less CPS (T6A[40%]) was avirulent and contrasted with the parent and corrected mutant, both of which caused sepsis in about half the mice.

We then analyzed the effect of CPS amount on the rate of intra-litter pup-to-pup transmission. This was carried out in the setting of influenza A co-infection administered to all pups in order to increase the frequency of transmission, thus allowing for meaningful comparisons [28]. Acquisition of infection by contact pups from *Spn* colonized index pups was proportional to the amount of CPS (Table 2). Transmission rates were significantly reduced for type 6A (40%) and type 4 (0 and 20%) mutants compared to strains expressing higher levels of CPS.

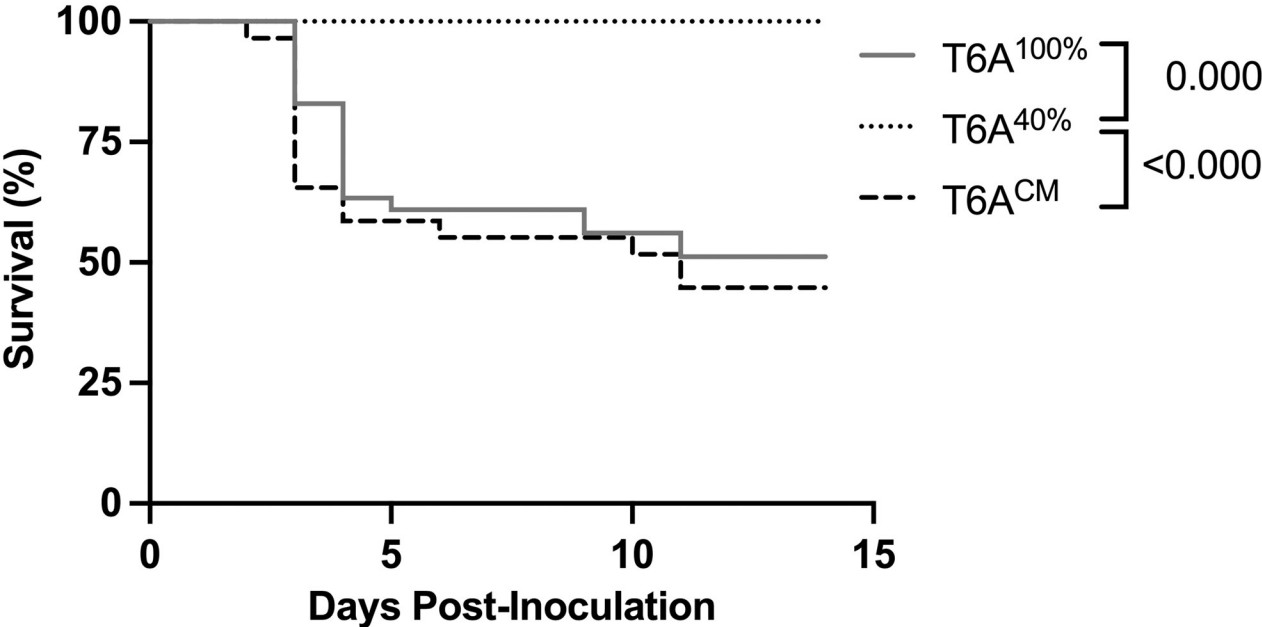

**Fig 5. CPS amount affects sepsis following colonization.** 10-day-old pups were challenged intranasally with T6A strains (~$10^2$ CFU) and survival monitored for 14 days. Statistical significance was determined using the Kaplan-Meier test followed by Dunn's correction. n = 22-41/group.

**Table 2. Effect of capsule polysaccharide amount on transmission.**

| Strain | Total Contacts | Colonized Contacts[#] | Transmission (%) | P-value (vs T6A[100%])[*] |
|---|---|---|---|---|
| T6A[100%] | 14 | 13 | 93 | - |
| T6A[40%] | 17 | 9 | 53 | 0.0207 |
| T6A[CM] | 15 | 14 | 93 | >0.9999 |
| Strain | Total Contacts | Colonized contacts[#] | Transmission (%) | P-value (vs T4[100%])[*] |
| T4[100%] | 12 | 11 | 92 | - |
| T4[50%] | 12 | 10 | 83 | >0.9999 |
| T4[20%] | 13 | 1 | 8 | <0.0001 |
| T4[0%] | 11 | 0 | 0 | <0.0001 |

[#]Colonization was assessed 7 days (T6A strains) or 9-10 days (T4 strains) post-pneumococcal inoculation of index pups.

[*]Fisher's Exact Test

## Comparison of CPS amount in clinical isolates

Lastly, we used immunoblotting to compare CPS amount among clinical isolates of the same type, but differing in date, site and location of collection. Nine type 6A isolates, including P376 analyzed above, varied by as much as 65-fold in the amount of CPS under standard growth conditions (Fig 6). For illustration purposes, both opaque (O) and transparent (T) forms of P461 were included and showed the expected difference of ~10-fold. These differences in CPS expression were not limited to type 6A. Isolates of two other common *Spn* types 6B (n = 11) and 23F (n = 10) also showed 35- and 6.6-fold variation, respectively, in amounts of CPS within a given type. This limited survey demonstrates that CPS amount is a highly variable feature of *Spn*.

To further explore effects of capsule amount during infection, one of the type 6A clinical isolates expressing a relatively low amount of capsule was selected. P592 was unable to cause bacteremia following IP inoculation at a low dose ($10^2$ CFU) as expected (Fig 7A). Following

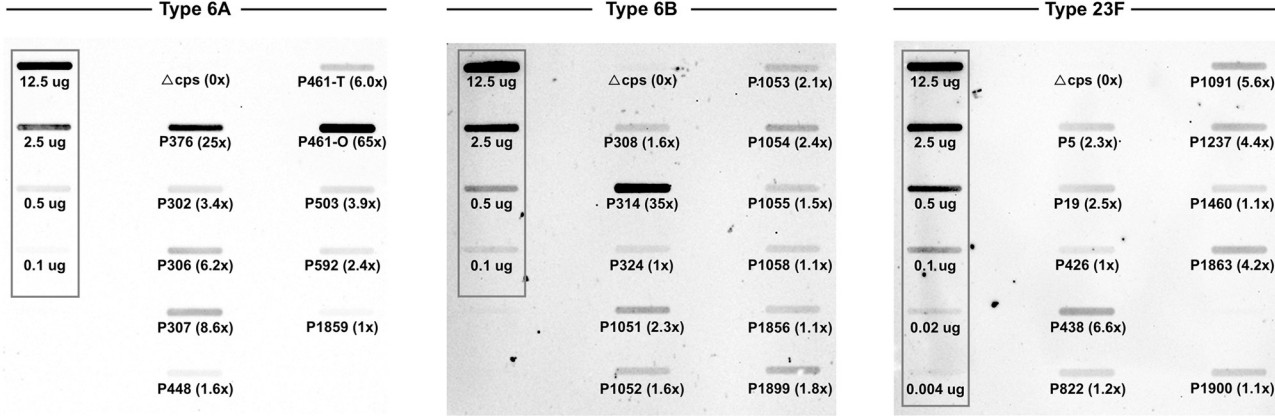

**Fig 6. Variation in CPS amount among clinical isolates.** Clinical isolates from our collection (labelled by strain P#) of three CPS types (6A, 6B, 23F) were grown to mid-log phase and a lysate adjusted for equal loading was applied to a nitrocellulose membrane. CPS was detected with type-specific antisera and relative binding compared in relation to a standard curve generated using purified CPS of the same type (shown boxed with amount loaded indicated) and quantified by densitometry. The level of CPS of each isolate is expressed relative to the strain of the same type with the least CPS (1x). An unencapsulated mutant served as negative control (Δcps).

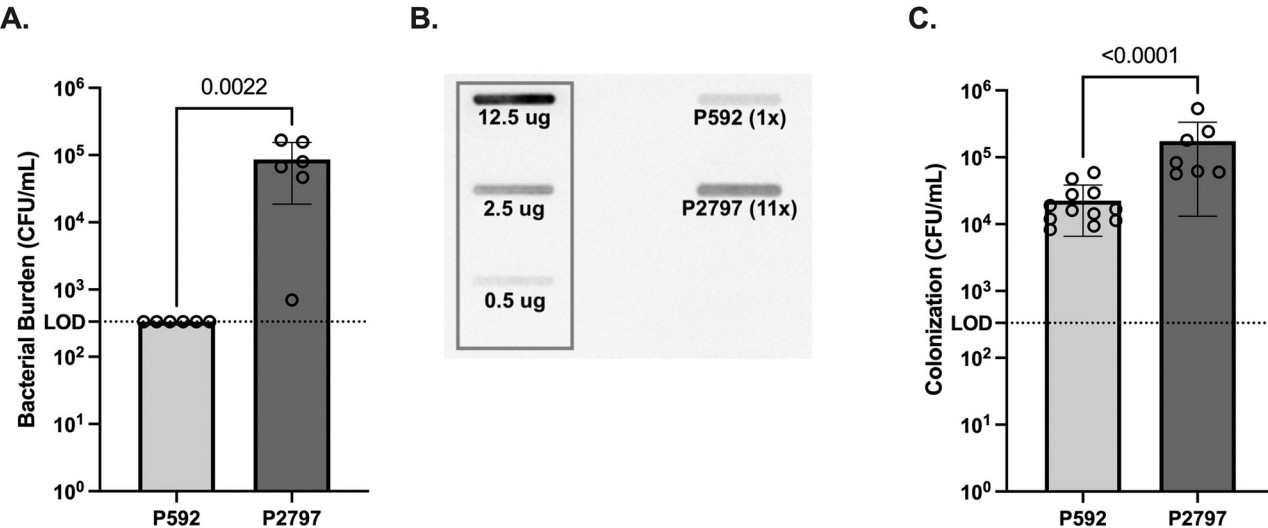

**Fig 7. Analysis of a clinical isolate.** (A)10-day-old pups were challenged intraperitoneally ($10^2$ CFU) and sacrificed 24 hrs later when blood was collected for quantitative culture. Strains included the clinical isolate P592 and P2797, an opaque variant obtained from the bloodstream of a pup challenged with P592 at a high dose. (B) Immunoblots of lysates with serotype 6A-specific anti-capsule antibody. Comparison to a standard curve generated using type 6A CPS was used to compare the relative amount of CPS in P2797 compared to the parent P592. (C) Pups at 10 days of age were challenged IN with $10^{2-3}$ CFU of the strain indicated. URT lavages were collected at 3 days post-infection for quantification of colonization density. Statistical significance was determined using the Mann-Whitney test.

IP challenge with a high dose ($10^7$ CFU), a variant P2797, with a colony phenotype of increased opacity, was obtained from blood culture of a single pup (Table 1). In contrast to its parent, P2797 was able to cause bacteremia following challenge at a low dose ($10^2$ CFU). Immunoblotting showed that P2797 expressed 11-fold more CPS relative to its parent (Fig 7B). Whole genome sequencing and comparison to P592 revealed that these changes were associated with acquisition of a Thr321Ala missense mutation in CpsE of P2797. P592 colonized pups at relatively low levels (Fig 7C). The selection for this mutation that increased CPS expression in P2797 allowed for more robust colonization. Together, these observations add to the evidence that capsule amounts permissive for invasive infection enhance colonization.

## Discussion

This study examined the effects of differences in amounts of the major virulence determinant of a leading pathogen. Defined, isogenic mutants carrying a single spontaneous mutation in the *cps* locus (T6A) or strains with changes to the *cps* promoter region (T4) were compared. Amounts of CPS per bacterial cell based on immunological assays with type-specific antibodies were used as a proxy for capsule thickness. Results confirm that thick capsules are required to inhibit i) binding of antibodies to underlying conserved surface antigens and ii) deposition of serum complement. These characteristics correlated with the capacity to evade opsonization, sustain bacteremic infection, and cause sepsis following invasion from the URT. However, the characteristics promoting invasive infection would not explain why *Spn* expresses thick capsule, as it is a 'dead end' for the organism. Thus, we investigated how CPS amount factors into the critical aspects of its commensal lifestyle; establishment in the URT (retention), stable colonization (density and duration), and transmission from host-to-host. CPS amount alone was shown to be important for each step in the dynamics of *Spn* colonization (retention, density and duration), which in turn impacts the chances of successful transmission. Although our

study focused exclusively on infant models, our findings could be relevant to infection at later ages, including in the elderly, who also have a high burden of *Spn* infection.

Several aspects of the mechanism involved in these observations merit further comment. WT T6A and T4 strains expressed amounts of CPS that effectively shielded the bacterial surface from potential opsonins and were permissive for bacteremic infection. Threshold amounts of CPS associated with these characteristics, however, appeared to differ between the two serotypes tested (>45% for T6A and <50% for T4). Thus, the relative importance of amounts of CPS must be considered separately for any given *Spn* type, since the physical properties and chemical composition of CPSs vary. In this regard, Magee and Yother previously examined the requirement for capsule during murine colonization and invasive infection [29]. They concluded that isolates producing reduced levels of capsule remain capable of colonization and causing invasive infection. Their study, however, depended on type 3 strains, which are unusual among pneumococci because they produce highly mucoid colonies. While type 3 strains, which express very thick capsules compared to other types, retain fitness in vivo with ~20% of WT levels of CPS, the findings here suggest this is generally not the case.

Additionally, we previously showed that unencapsulated *Spn* demonstrate an impaired ability to escape mucus binding within the loose mucus layer during the first minutes after arrival in the URT [7]. This finding indicated that capsule could inhibit initial clearance which occurs by the mechanical removal of particles embedded in loose mucus by mucociliary flow. Entrapment in mucus, which is also facilitated by the agglutinating effect of specific mucosal antibody, prevents establishment along the epithelial surface where stable colonization occurs [30]. Results in this report show that similar to unencapsulated mutants, strains expressing reduced amounts of CPS are also more susceptible to rapid removal, as seen with decreased initial retention in the URT [7]. This effect correlates with previous findings using the T4 strains analyzed here showing that mucus binding in a plate assay inversely correlates with amounts CPS–a result that could be explained by repulsive hydrostatic and charge properties of CPS [22]. A thicker capsule containing more CPS, therefore, appears to enhance mucus evasion and could explain why it is advantageous despite the metabolic cost for its biosynthesis. This also suggests that CPS-mediated evasion of mucus offsets any negative effects of capsule in adhering to host cells as seen in tissue culture models [13].

Early success in establishment on the URT mucosal surface has several consequences for the organism. We have shown that the *Spn* that initially colonize their niche have a substantial competitive advantage over those arriving later due to the early production of quorum sensing-dependent anti-*Spn* fratricidal effectors [24]. Additionally, when comparing isogenic constructs of different serotypes (capsule-switch mutants), we observed that a competitive success during the first hours of establishment in the host correlated with colonization duration over many weeks [26].

Similarly, the effect of increased CPS amount in facilitating initial success in the URT correlated with long term persistence. The effect of CPS amount on sustaining high colonization density and in prolonging colonization duration was shown here to be dependent on IL-17 signaling. This could be explained by IL-17-dependent recruitment of professional phagocytes [25]. Thicker capsules could inhibit the non-opsonic uptake by immune cells by macrophage scavenger receptors such as MARCO during colonization [31]. It should also be noted that in mouse studies neither complement nor specific antibodies generated during colonization appear to contribute to URT clearance once colonization is established [32,33]. This provides further evidence that the capsule-mediated shielding against the deposition of opsonins may not be the capsule's primary function during commensal host interactions. This is in contrast to invasive infection, where the effect of capsule amount in blocking the deposition of complement correlated with *Spn's* ability to survive in the bloodstream. For those pneumococcal types

cleared from the bloodstream in a complement-independent manner, it would be interesting to examine the effect of capsule amount on their direct interaction with phagocytes, including liver-resident macrophage Kupffer cells [34].

Increased CPS amount also correlated with the rate of transmission from pup-to-pup. We previously showed that more CPS per strain enhances T4 *Spn* shedding, a rate limiting step for transmission in this model [22]. Shedding was not considered separately in this report, since colonization density was affected by capsule amount and is a main determinant of the number of shed *Spn*. Transmission also requires robust colonization. Since CPS amount affected colonization density over the time period during which contact and index pups were together, we cannot distinguish an independent effect on transmission apart from that on colonization. Also, because of differences in colonization dynamics, we could not distinguish whether differences in transmission were due to altered shedding from the index pups or acquisition and retention by contact pups.

The isogenic mutants tested here varied up to 5-fold (20% to 100%) of WT amounts of CPS. When clinical isolates of three serotypes were compared, we observed a far greater range of CPS amounts (up to 65-fold). Some of these differences could be due to opacity phase variation, which previously was reported to account for up to a 5.6-fold difference in capsule amount [21]. Many of the type 6A clinical isolates expressed less than 40% of the level CPS of P376 (T6A$^{100\%}$), which in the animal model was well below the threshold amount required for invasive infection, robust colonization, and efficient transmission. This wide range of variation was not unexpected considering variations in colony morphology, ranging from rough to mucoid, among different isolates. It is now clear that multiple factors contribute to the variation in CPS expression among isolates, including opacity phase variation, differences in *cps* promoter strength, a *cis*-acting regulatory element, multiple transcriptional factors, translational control by environmental conditions such as temperature, and post-translational control by tyrosine phosphorylation of CpsD [21,35–39]. As the current report shows that much smaller differences in CPS amounts (<5-fold) affect each aspect of *Spn* host interactions, variation in this bacterial factor should be taken into account when comparing the pathogenicity of different isolates.

Finally, since many of the bacterial species found on mucosal surfaces express encapsulated forms, our findings demonstrating how CPS amount affects colonization and transmission dynamics may be relevant to other species.

## Materials and methods

### Ethics statement

All animal experiments followed the guidelines summarized by the National Science Foundation Animal Welfare Act (AWA) and the Public Health Service Policy on the Humane Care and Use of Laboratory Animals. The Institutional Animal Care and Use Committee (IACUC) at New York University Grossman School of Medicine oversees the welfare, well-being, proper care and use of all animals, and they have approved the protocol used in this study (IA16-00538).

C57BL/6J WT mice were purchased from The Jackson Laboratory (Bar Harbor, ME) and congenic *IL17ra$^{-/-}$* knock-out mice were acquired from Amgen Inc. Mice were bred and housed in a conventional animal facility. Pups were maintained with their dam until weaning at age 3 wk. Weaned mice were fed ad lib the PicoLab Rodent Diet 20, a 20% protein diet formulation, and were given acidified water for consumption. Additionally, the animals were kept on a light-cycle of 12 hours on, 12 hours off with a temperature in the animal facility of 70˚F (±2˚F). Pups were monitored daily for signs of sepsis.

## Bacterial culture and strain construction

Types 4 and 6A strains used in the study are described in Table 1. Clinical isolates were confirmed as *Spn* by sensitivity to ethylhydrocupreine hydrochloride (optochin) and typed using specific antisera (Statens Serum Institute). Pneumococci were grown statically in tryptic soy (TS) broth (Becton, Dickinson) at 37˚C. Upon reaching the desired $OD_{620nm}$ of ~1.0, cells were washed and diluted in sterile phosphate-buffered saline (PBS) for inoculation. For quantitative culture, serial dilutions were plated on TS broth agar supplemented with an appropriate antibiotic and either 5% sheep blood or catalase (6,300 U/plate; Worthington Biochemical Corporation) and incubated overnight at 37˚C with 5% $CO_2$.

Colonies expressing different amounts of CPS were selected by visual screening using microscopy with oblique, transmitted illumination as described [16]. Mutants were confirmed by whole-genome sequencing. A corrected mutant (P2752) of the *cps*E mutation in P385 was constructed by transforming P385 as previously described [26] with the *cps* PCR product of P376 using primers 5'- ACC ATT GTC TCT ACC TCT CAC -3' and 5'- CGG AAT TCC TGT AAT TGA TGT CAT -3'. The transformation was verified using sequencing primer 5'- GAA GAT TCT CCT ACT TAC AGC AAC -3.

## Whole-genome sequencing and genomic analysis

DNA libraries for strains P376, P384, and P385 were sequenced on an Illumina NovaSeq 6000 instrument to a coverage of at least 560x, producing 150 bp paired reads. The software fastp v0.20.1 was used with default settings to trim low-quality bases, remove adapters, and filter out low-quality reads [40]. The resulting trimmed, filtered reads of each isolate were aligned to an annotated genome assembly of *Spn* strain 6A-10 (RefSeq accession number: GCF_013047165.1) using Snippy v4.6.0 (https://github.com/tseemann/snippy/). The Snippy command snippy-core was used to compute a core-genome alignment and to call single-nucleotide variants (SNVs). The resulting core genome alignment had 2,007,997 positions, of which 3,072 were variable.

Accession number(s): The sequencing reads generated for this study are available in NCBI under BioProject PRJNA930766.

## Immunoblotting

To visualize the differences in CPS expression, strains were grown to $OD_{620nm}$ of 1.0 and an aliquot saved for total protein determination. 1 mL of culture was collected and then resuspended in PBS before lysis with 0.2% Triton-X 100 at RT. To 180 μL of lysed cells, 2 μL of proteinase K (50ug/ul) and 28 μL of 1X PBS were added to degrade cellular proteins at 65˚C for 15 min. The samples were then diluted 1:100 in 1X PBS and sonicated for homogenization at 60 Hz for 6x10 seconds. To perform the blot, a vacuum-slot blot device was used with sample loading adjusted based on bacterial density or total protein as indicated. Samples were loaded onto a nitrocellulose membrane alongside a purified CPS standard (Merck) of the same *Spn* serotype. The membrane was blocked in 5% milk in PBST for 30 min and then incubated with type-specific rabbit antiserum (Statens Serum Institute, 1:8000 to 40000) for 1 hr at RT. An unencapsulated strain, P2422, was incubated with type-specific antiserum to absorb out non-CPS specific binding. After washing the membrane, it was incubated with HRP-conjugated IgG goat anti-rabbit antibody (1:5000) for 1 hr at RT. The membrane was then developed with Thermo Femto Super sensitivity substrate and imaged using the iBright imaging system. Chemifluorescence readings were processed and densitometry area-under-the-curve analysis were performed in FIJI.

## Bacteremia model

10-day-old pups of both sexes were given an IP inoculation containing $10^2$ CFU of *Spn* suspended in 40 µl of PBS, as described previously [27]. Pups were euthanized 24 hrs post-inoculation, and blood was collected via cardiac puncture and cultured to assess for bacteremia. To select for a variant expressing increased CPS, the inoculum was increased to $10^7$ CFU. To verify its importance in invasive infection, complement was depleted in 9-day-old pups of both sexes by IP administration of 20 ug of cobra venom factor (or PBS as a vehicle control) (Sigma-Aldrich) in 20 µL in PBS. The pups were then inoculated with *Spn* in the same manner as described above 24 hrs later. At 24 hrs post-inoculation, the pups were euthanized to perform blood collection and culture.

## Colonization model

Pups of both sexes were given an intranasal inoculation without anesthesia containing $10^2$ CFU of *Spn* suspended in 3 µl of PBS, as described previously [41] at either 4 days of age (T4 strains) or 10 days of age (T6A strains). To measure colonization density, pups were euthanized at the indicated time point by $CO_2$ asphyxiation followed by cardiac puncture. The URT was lavaged with 250–500 µl of sterile PBS from a needle inserted into the trachea, and fluid was collected from the nares for quantitative culture. For strains not resistant to streptomycin (200 ug/ml), neomycin (5 ug/ml) was used to select against contaminants.

To assess *Spn* retention, pups were euthanized at 4 hrs post-infection following an inoculum of $10^{2-3}$ CFU. To evaluate *in vivo* competition at 4 hrs, pups were co-infected with two strains expressing different amounts of CPS ($\sim 10^{2-3}$ CFU per strain). Pups were euthanized at 4 hrs post-infection, and URT lavages were collected and plated on TS agar plates with catalase and appropriate antibiotics. Numbers of each strain were obtained based on morphological differences in colonies. Competition indices were calculated as the ratio of strains in the output divided by the ratio of strains in the inoculum.

To compare invasive infection following intranasal challenge, pups were inoculated intranasally at the dose and age described above. Pups were monitored daily for survival or moribund endpoint for 14 days following inoculation.

## Comparison of surface shielding

Bacterial strains grown in TS broth at 37˚C to an $OD_{620nm}$ of 1.0 were washed and diluted in 1% PBS-BSA to a final concentration of $10^6$ CFU in 100 µL. Bacteria were stained with the mAb to phosphorylcholine, TEPC-15 (Sigma-Aldrich, 1:5000 dilution) followed by a FITC-labeled antibody to murine IgA. Reactions were carried out at 4˚ C for 30 mins without spin or wash steps in between. After staining, samples were fixed with 4% paraformaldehyde (PFA), washed once with PBS, and resuspended in PBS. The surface shielding assay was also performed using *ex vivo* samples of bacteria collected from colonized mice. T6A strains P376 or P385 were administered IN to 10-day-old pups at a dose of $10^6$ CFU in 3 µL. Two hours post-inoculation, pups were euthanized, and URT lavages collected. Lavages were pooled from at least 6 pups, pelleted and resuspended in 100 µL and treated as described above. Samples were analyzed by flow cytometry using the LSRII flow cytometer (BD Biosciences) and analyzed using FlowJo software (Tree Star).

## Comparison of complement binding

Bacterial strains grown in TS broth at 37˚C to an OD of 1.0 at 620nm were washed and resuspended in 1X PBS. $2.5 \times 10^6$ CFU of *Spn* in 25 µL were incubated with an equal volume of fresh

normal mouse serum at 37 ˚C for 30 min. For negative controls, serum was pre-treated at 57˚C for 30 min to heat-inactivate complement. The incubation was ended by adding 50 μL of 20 mM EDTA in 1X PBS. Cells were pelleted, resuspended, and then stained with FITC-conjugated anti-mouse C3 antibody (MP Biomedicals) at 4˚C for 30 min. Cells were then fixed in 4% PFA and resuspended in 20 mM EDTA in 1x PBS. Samples were analyzed by flow cytometry using the LSRII flow cytometer (BD Biosciences) and analyzed using FlowJo software (Tree Star).

### Transmission model

The pneumococcal transmission model with influenza A (IAV) co-infection was described in previous studies [42]. Briefly, one in four to five pups in the litter was randomly selected and, at an age of 4 days, infected with *Spn* (index mice) and then returned to cage containing the dam and the other uninfected pups (contact mice). For experiments involving type 6A strains, all pups in the litter were inoculated intranasally with 3 μl IAV/HKx31 containing 250 plaque-forming units three days after *Spn* infection. For experiments with type 4 strains, all pups were inoculated with IAV four days after *Spn* infection. Then, all pups were euthanized and nasal lavages were collected either four days after IAV infection (T6A strains) or six days after IAV infection (T4 strains). Lavages were cultured to detect bacterial transmission from the index to contact pups.

### Statistical analysis

All statistical analyses were performed using GraphPad Prism 9.2.0 (GraphPad Software, Inc., San Diego, CA).

### Dryad DOI

https://datadryad.org/stash/share/jpNBGQpWrHUawsaKV3Gen_3qFoVifuW5JqxMX-1xVd8 [43].

## Author Contributions

**Conceptualization:** Annie R. Abruzzo, Gavyn Chern Wei Bee, Gregory Putzel, Surya D. Aggarwal, Hannes Eichner, Jeffrey N. Weiser.

**Data curation:** Jiaqi Zhu, Annie R. Abruzzo, Cindy Wu, Gavyn Chern Wei Bee, Alejandro Pironti, Gregory Putzel, Surya D. Aggarwal, Hannes Eichner.

**Formal analysis:** Jiaqi Zhu, Annie R. Abruzzo, Cindy Wu, Gavyn Chern Wei Bee, Alejandro Pironti, Gregory Putzel, Surya D. Aggarwal, Hannes Eichner, Jeffrey N. Weiser.

**Funding acquisition:** Jeffrey N. Weiser.

**Investigation:** Jiaqi Zhu, Annie R. Abruzzo, Cindy Wu, Surya D. Aggarwal.

**Methodology:** Jiaqi Zhu, Annie R. Abruzzo, Cindy Wu, Gavyn Chern Wei Bee, Gregory Putzel, Surya D. Aggarwal, Hannes Eichner.

**Project administration:** Jeffrey N. Weiser.

**Software:** Alejandro Pironti, Gregory Putzel.

**Supervision:** Alejandro Pironti, Jeffrey N. Weiser.

**Validation:** Jiaqi Zhu, Surya D. Aggarwal.

**Writing – original draft:** Jeffrey N. Weiser.

**Writing – review & editing:** Jiaqi Zhu, Annie R. Abruzzo, Cindy Wu, Alejandro Pironti, Gregory Putzel, Surya D. Aggarwal, Hannes Eichner, Jeffrey N. Weiser.

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
