## [Decision Letter · Decision Letter 0]

16 Apr 2023

Dear Jeff,

Thank you very much for submitting your manuscript "Effects of Capsular Polysaccharide Amount on Pneumococcal-Host Interactions" for consideration at PLOS Pathogens. As with all papers reviewed by the journal, your manuscript was reviewed by members of the editorial board and by several independent reviewers. The reviewers appreciated the attention to an important topic. Based on the reviews, we are likely to accept this manuscript for publication, providing that you address the review recommendations. For example, reviewer 3 suggested that insight into or comment on possible mechanism(s) by which capsule might promote colonization (beyond counteracting an IL-17-dependent mechanism) or bacteremia (beyond counteracting complement; e.g., see Kupffer cell interaction, PMID 35258522) would strengthen this interesting manuscript. Please also note that since the submission of this manuscript, an analysis of different capsular serotypes has been published, PMID 36943879, that may be appropriate to include in your comments.

Sincerely,

John

John M Leong

Pearls Editor

PLOS Pathogens

Marcel Behr

Section Editor

PLOS Pathogens

Kasturi Haldar

Editor-in-Chief

PLOS Pathogens

orcid.org/0000-0001-5065-158X

Michael Malim

Editor-in-Chief

PLOS Pathogens

orcid.org/0000-0002-7699-2064

Reviewer Comments (if any, and for reference):

Reviewer's Responses to Questions

**Part I - Summary**

Reviewer #1: This is a well written, concise manuscript examining the functional impact of variations in capsule thickness on steps in pathogenesis. The question is clearly posed in the context of opposing benefits in sites such as nasopharynx and blood. More capsule is believed to decrease interactions with host cells which impairs colonization and invasion but at the same time aid colonization and invasion by preventing phagocytosis, complement deposition and mucous clearance. This study has answers to this conundrum and indicates that capsule amount has a clear cutoff of benefit in each of several steps in pathogenesis.

The shielding assay is an excellent readout for capsule function.

Each assay is clearly explained and graphs show distinct differences.

The final graphs provide a nice peak into the clinical realm for relevance.

Reviewer #2: Streptococcus pneumoniae, part of the normal human nasopharyngeal microflora, is also a major pathogen whose capacity to cause invasive disease is critically dependent on production of a polysaccharide capsule, of which multiple serotypes exist. Invasive disease is invariably preceded by colonization of the upper respiratory tract, and such colonization also acts as a reservoir for transmission to new hosts. The critical role of the capsule in invasive disease is well understood, but its contribution to colonization and transmission has received only minimal attention to date. These early events are of fundamental importance to the host-pathogen interaction, particularly given that progression to invasive disease is an evolutionary dead end. The total amount of capsule presented on the pneumococcal surface impacts the degree of exposure of other surface components important for colonisation (e.g. adhesins), and the pneumococcus is known to be capable of regulating capsular expression by a variety of mechanisms. The generally held view has been that capsule might be less important in the nasopharynx than in the blood, where the invading organism is subject to highly efficient reticuloendothelial clearance mechanisms.

The present study turns this assumption on its head, providing convincing evidence that capacity to both colonise the nasopharynx in infant mice and to transmit to new hosts depends on maximal capsule expression. The authors use mutants in 2 different serotype backgrounds (4 and 6A) with reduced cps expression (due to point mutations in cpsE) and a genetically repaired fully encapsulated strain, to demonstrate that strains with <50% of the wild type level of cps expression exhibit poorer bacterial retention in the nasopharynx as well as more efficient clearance later in colonisation. This correlated with reduced transmission to new hosts. The strains with reduced cps expression exhibited greater accessibility of deeper cell wall structures to exogenous antibody and greater C3 deposition on the cell surface when exposed to normal mouse serum. Interestingly, differences in colonization were not seen in IL-17ra-/- mice, implying an important role for innate signalling in recruitment of neutrophils to facilitate bacterial clearance. The study provides important new insights into these critical events in the host-pathogen interaction, that are potentially applicable to a wider range of bacterial pathogens. The experiments are well planned and executed and the data are presented in a clear and logical fashion.

Reviewer #3: Strength

This paper describes the impact of pneumococcal capsule on colonization, transmission and invasive infection. The strengths include: 1) the importance of understanding the biology of capsule in pneumococcal carriage and invasive infection; 2) impact of capsule in shielding bacterial surface features; 3) using isogenic strains from two strain backgrounds for studying capsule.

Weakness

The paper is mostly descriptive in the nature of investigation from capsule amount, bacteremia in invasive infection, colonization, transmission and capsule thickness in wild strains. It lacks sufficient amount of data to explain how the amount of capsule makes those differences in various aspects of bacterial life/infection. In addition, the statement that the blood is the “dead end” for S. pneumoniae, and therefore thick capsule is meaningless for bacterial life and evolution seems to be too strong because bacteria in the blood may enhance bacterial transmission once being seeded into the lung environment from the circulation during bacteremic pneumonia. Moreover, the authors mostly cited their own work but did not make necessary connection to the previous studies in capsule function (see below).

**Part II – Major Issues: Key Experiments Required for Acceptance**

Reviewer #1: (No Response)

Reviewer #2: NIL

Reviewer #3: Major experiments

1. How does capsule thickness impact invasive infection? Can this be due to the impaired resistance of partial capsule to Kupffer cell capture in the liver? Bacterial burden in the blood, liver and spleen at various time points post ip inoculation should be very revealing.

2. Fig. 2B shows the importance of capsule in dealing with Th17-based immunity during colonization, how could this happen?

3. Fig. 4 shows variable thickness of capsule in wild strains. Can these variations impact pneumococcal colonization, transmission and invasive infection?

**Part III – Minor Issues: Editorial and Data Presentation Modifications**

Reviewer #1: Suggestions:

In view of the emphasis on amount of capsule, one area would be beneficial to further discuss. The spontaneous mutants are depicted as having a set amount of capsule. Yet it is likely that they still undergo processes such as phase variation and capsule shedding. Is this the case or are these more like the phase locked mutants developed in this lab? Either way, do the authors think that a 40% decrease in capsule is in the same ballpark as the changes invoked by phase variation or shedding? Or are the physiologic processes much less impactful on capsule amounts overall and represent just fine tuning? This might be assessed using the shielding assay.

Note: It would be helpful to define CM in the first Figure legend.

Reviewer #2: A minor irritation is the fact that the authors cite a couple of figure panels out of sequence (e.g. 1B cited after 1C, and 2E cited after figs 3 and 4). This may have been done to economise on the number of figures overall, but it doesn't help the flow of the paper. Perhaps the authors could consider whether some reorganisation (either text or figures) is in order at the revision stage.

Table 2 and lines 189-192: The reduced transmission to new hosts in the cps mutant immediately raises the question as to whether this is a consequence of reduced shedding by the index pups, or reduced establishment of colonization by the contact pups, or both. If the authors have data on levels of shedding then it would be good to present it. The point is raised later in the Discussion (lines 266-269), but perhaps it could be addressed at the first opportunity. It would also be helpful to have the time at which contacts were tested for colonisation to be specified in a footnote to Table 2, rather than having to refer to the Methods.

Line 474: there is a typo in the CFU/strain dose.

Reviewer #3: Page 19 line 383, assessment of surface shielding with TEPC-15 should cite a reference or provide justification for the method.

Page 19 line 388, full spelling of PFA should be provided.

Page 19 line 385, 391, “100uL” and “3uL” should be corrected with a space between number and unit; Greek letter should be used for micro.

Some importance literature needs to be cited to connect this work to the rest of the world, such as Magee et al. (doi: 10.1128/IAI.69.6.3755-3761.2001) has described pneumococcal strain expressing ~20% of the WT capsule is sufficient for nasal colonization; An et al. (doi: 10.1084/jem.20212032) report the importance of pneumococcal capsule in evading Kupffer cell capture in the liver in the context of invasive infection.

Figure 2, labeling of Y axis needs to be consistent across AB (bacterial burden) and CD (Bacteremia) if the two panels are talking about blood bacteria. Were blood bacteria in CD collected 24 hr post infection as in AB?

Being consistent with the expression of the same things across the paper, such as 100 CFU vs 102 CFU in Fig 2 and other figures.

Many more typo errors in the text, figure legends and references should be carefully corrected.

PLOS authors have the option to publish the peer review history of their article (what does this mean?). If published, this will include your full peer review and any attached files.

Reviewer #1: **Yes: **Elaine Tuomanen

Reviewer #2: No

Reviewer #3: No

Figure Files:

Data Requirements:

Reproducibility:

References:

---

## [Decision Letter · Decision Letter 1]

24 Jun 2023

Dear Dr. Weiser,

We are pleased to inform you that your manuscript 'Effects of Capsular Polysaccharide Amount on Pneumococcal-Host Interactions' has been provisionally accepted for publication in PLOS Pathogens.

Best regards,

John M Leong

Pearls Editor

PLOS Pathogens

Marcel Behr

Section Editor

PLOS Pathogens

Kasturi Haldar

Editor-in-Chief

PLOS Pathogens

orcid.org/0000-0001-5065-158X

Michael Malim

Editor-in-Chief

PLOS Pathogens

orcid.org/0000-0002-7699-2064

Reviewer Comments (if any, and for reference):

Reviewer's Responses to Questions

**Part I - Summary**

Reviewer #1: My suggestions were addressed.

Reviewer #2: See previous report

Reviewer #3: No

**Part II – Major Issues: Key Experiments Required for Acceptance**

Reviewer #1: I had none

Reviewer #2: Nil

Reviewer #3: No

**Part III – Minor Issues: Editorial and Data Presentation Modifications**

Reviewer #1: Addressed

Reviewer #2: The authors have satisfactorily addressed the minor concerns I had with the original version of this interesting and important manuscript.

Reviewer #3: No

PLOS authors have the option to publish the peer review history of their article (what does this mean?). If published, this will include your full peer review and any attached files.

Reviewer #1: **Yes: **Elaine Tuomanen

Reviewer #2: No

Reviewer #3: No

---

## [Editor Report · Acceptance letter]

1 Aug 2023

Dear Dr. Weiser,

We are delighted to inform you that your manuscript, "Effects of Capsular Polysaccharide Amount on Pneumococcal-Host Interactions," has been formally accepted for publication in PLOS Pathogens.

Best regards,

Kasturi Haldar

Editor-in-Chief

PLOS Pathogens

orcid.org/0000-0001-5065-158X

Michael Malim

Editor-in-Chief

PLOS Pathogens

orcid.org/0000-0002-7699-2064